# Characteristics of inclusions in chang 7 member and shale oil accumulation stages in Zhijing-Ansai area, Ordos Basin

**Wenjie Wu, Jian Wang\*, Nan Wu, Yong Feng, Yilin Liang, Yulin Chen**

College of Geosciences, Yangtze University, Hubei, Wuhan, 430100, China

\* wangjian2017@yangtzeu.edu.cn

**Data Availability Statement:** All relevant data are within the manuscript and its Supporting Information files.

## Abstract

In order to further clarify the shale oil accumulation period of the Chang 7 member of the Mesozoic Triassic Yanchang Formation in the Zhijing-Ansai area of the central Ordos Basin, Using fluid inclusion petrography analysis, microscopic temperature measurement, salinity analysis and fluorescence spectrum analysis methods, combined with the burial history-thermal history recovery in the area, the oil and gas accumulation period of the Chang 7 member of the Yanchang Formation in the Zhijing-Ansai area was comprehensively analyzed. Sixteen shale oil reservoir samples of the Mesozoic Triassic Yanchang Formation in seven typical wells in the study area were selected. The results show that the fluid inclusions in the Chang 7 member of Yanchang Formation can be divided into two stages. The first stage inclusions mainly develop liquid hydrocarbon inclusions and a large number of associated brine inclusions, which are mainly beaded in fracture-filled quartz and fracture-filled calcite. The fluorescence color is blue and blue-green, and the homogenization temperature of the associated brine inclusions is between 90–110°C. The second stage inclusions are mainly gas-liquid two-phase hydrocarbon inclusions, gas inclusions and asphalt inclusions. Most of them are distributed in the fracture-filled quartz, and the temperature of the associated brine inclusions is between 120–130°C. Fluid inclusions in Chang 7 member of the Yanchang Formation can be divided into two stages. The $CO_2$ inclusions and high temperature inclusions in the Chang 7 member of the Yanchang Formation are mainly derived from deep volcanic activity in the crust.

## Introduction

In the past few decades, with the continuous decline of conventional oil and gas resources, global oil and gas resources are facing great challenges, and the exploration and development of shale oil has always been highly valued by experts and scholars [1].

In the Ordos Basin, the Chang 7 member of the Mesozoic Triassic Yanchang Formation is an important shale oil reservoir with great potential for exploration and development. The Yanchang Formation has experienced many tectonic movements, and the accumulation conditions are extremely complex [2]. Predecessors have done a lot of research work on the oil

**Funding:** This work was supported by Key Laboratory of Tectonics and Petroleum Resources (China University of Geosciences), Ministry of Education (Grant number TPR-2022-12), and the National Nature Science Foundation of China (Grant numbers 42172179). The project fund is hosted by Professor Nan,Wu.

**Competing interests:** The authors have declared that no competing interests exist.

and gas accumulation periods of the Yanchang Formation. Currently, there are three different viewpoints in the research on the oil and gas accumulation period of the Chang 7: (1) The first stage of charging and accumulation occurred at the end of the Early Cretaceous [3–5]. Some scholars believe that this period is the peak period of basin subsidence and thermal evolution, during which oil and gas accumulation occurred [6]. (2) Two stages of charging and accumulation. One was small-scale oil and gas charging in the Early Cretaceous [7,8], and the other was large-scale oil and gas accumulation in the late Late Cretaceous [9–11]. (3) Three phases of charging and accumulation. The first stage of oil and gas charging occurred at the end of the Middle Jurassic [12–14], and the second stage of large-scale oil and gas migration occurred in the Late Jurassic to the middle of the Early Cretaceous [15–17]. The three phases of oil and gas accumulation occurred in the Late Cretaceous [18,19]. Due to the different understandings of the periods of oil and gas accumulation by the predecessors, the research on the key period of the accumulation age in this area needs to be perfected, and there is a lack of systematic understanding of the oil and gas accumulation in this block. Therefore, it is necessary to conduct further research on the oil and gas accumulation periods of the Chang 7 of the Yanchang Formation in the Zhijing Ansai area of the Ordos Basin.

This research project focused on the Chang 7, carried out experiments through petrographic observation of fluid inclusions, composition analysis, temperature and salt test, to identify the oil and gas accumulation periods of the Chang 7. In this paper, on the basis of systematic sampling and research on the characteristics of the low-permeability sandstone reservoirs of the Mesozoic Yanchang Formation in the Zhijing-Ansai area [20], the microscopic petrographic observation of fluid inclusions, microthermometry characteristics, and burial and thermal history are used to discuss. The stages of oil and gas migration and charging in the Mesozoic Yanchang Formation and the determination of reservoir formation time [21]. The research results help to understand the oil and gas accumulation mechanism of the Yanchang Formation in the Zhijing Ansai area of the Ordos Basin, and can provide theoretical support for the next step of oil and gas exploration in favorable areas in this area.

## Geological background

The Ordos Basin is located in the western part of the North China Plate, spanning five provinces of Shanxi, Gansu, Ningxia, Mongolia, and Shanxi, with an area of about 370,000 square kilometers [22]. The geological background of the Ordos Basin is complex, and its tectonic activities include six first-order tectonic units: Yimeng Uplift, West Margin Thrust Zone, Tianhuan Sag, Yishan Slope, Jinxi Flexure Belt, and Weibei Uplift [23,24]. The period of the Chang 7 member of the Triassic Yanchang Formation in the Ordos Basin was the heyday of the lake basin, and the Late Triassic was the most important stage for the development of geological structures in the basin, and it was also one of the periods when the oil and gas system was extremely developed (Fig 1). The Chang 7 member in the Ordos Basin belongs to the heyday of basin development and evolution. The complex geological history and a series of sedimentary structures make the Chang 7 member the most important source rock development layer and key productivity layer in the Ordos Basin [25,26].

The research area is the Zhijing-Ansai area of the Ordos Basin, and the Chang 7 member oil layer group in the research area can be divided into three sublayers, Chang $7_1$, Chang $7_2$ and Chang $7_3$ according to the depositional cycle from top to bottom. The lithology of the reservoir is mainly feldspathic sandstone and lithic sandstone, with high content of clay minerals, the reservoir is compact, and intergranular dissolution pores and intragranular dissolution pores are developed. The Chang 7 member mainly develops delta front subfacies and shore-shallow lake subfacies, partially develops semi-deep lake subfacies, and delta front mainly

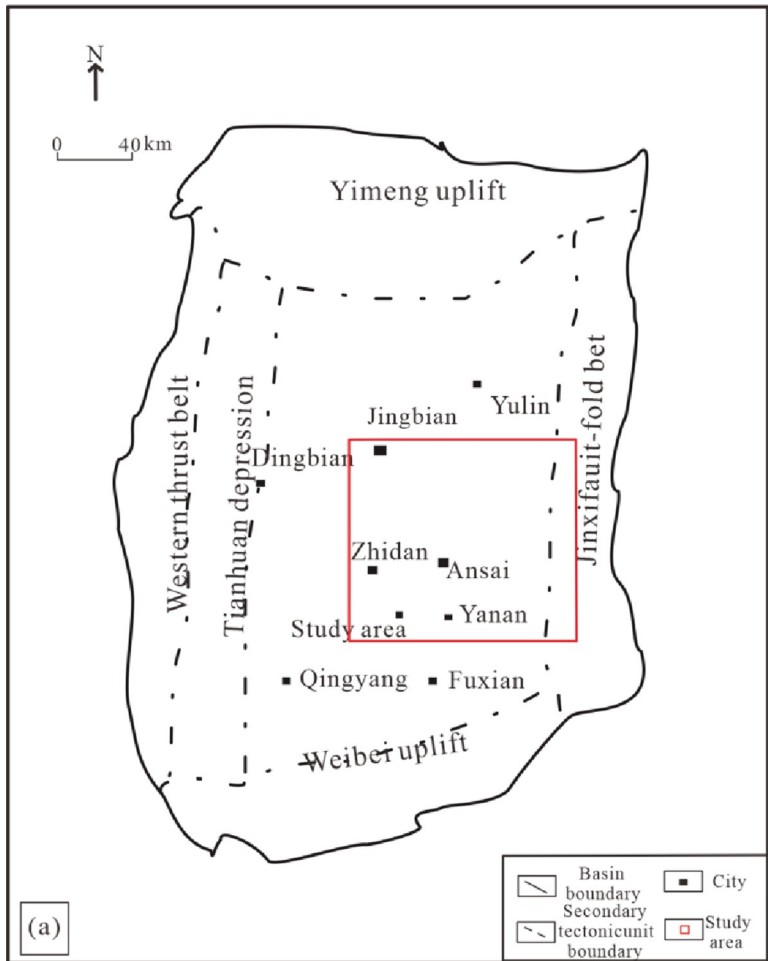

**Fig 1. The main tectonic units and the location of the study area in the schematic diagram of Ordos Basin.**
(Modified from Liu et al., 2015) [27] (https://www.cia.gov/the-world-factbook/countries/china/map).

develops underwater distributary channels, flow separately bays and mouth bar microfacies
[28].

## Materials and methods

### Samples and equipment

The samples collected for this study were collected from 7 typical wells in the Zhijing Ansai
Block, including Well Xin 140, Well Xin 283, Well Shun 37, Well Shun 111, Well Gao 135,
Well Wu 100, and Well Qiao 136. Select 16 samples of Chang 7 member for preparation.
Under the indoor conditions of room temperature 20°C and air humidity 30%, make double-
sided polished thin slices, and further carry out petrographic observation, composition analy-
sis and temperature and salt test of fluid inclusions. The equipment used in this experiment is
a Nikon Eclipse 80i dual-channel fluorescence microscope equipped with ultraviolet (UV) and
transmitted light (TR). The excitation wavelength of ultraviolet light is 330-380nm, and the
micro fluorescence spectrometer adopts the Maya 2000 Pro fiber optic spectrometer in the
United States.

### Fluid inclusion analysis

The composition of fluid inclusions represents the original mineral composition of past geological periods. The formation of inclusions runs through the entire geological process. It records and preserves the physicochemical characteristics of different stages of geological processes [29]. By describing the morphology and spatial distribution characteristics of inclusions under fluorescence and transmitted light by microscopy, oil and gas inclusions and brine inclusions can be identified. Under the excitation of ultraviolet light, hydrocarbon inclusions generally emit various fluorescences, and the organic molecular structure in the inclusions is the main factor controlling the fluorescence intensity and color [30]. Oil inclusions with different fluorescent colors represent different maturity. Using Spectra Suite software to calculate the spectral parameters of hydrocarbon-bearing inclusions one by one, the maturity of individual inclusions can be judged.

The liquid hydrocarbon inclusions and gas-liquid two-phase brine inclusions with different occurrences in different periods were selected, and their homogeneous temperatures were measured respectively, and the measured brine temperature in the same period was used to approximate the geothermal temperature when the oil inclusions were trapped.

Temperature measurement and salt measurement experiments were carried out under transmitted light, using the British Linkam THMSG600 microscopic cold-hot stage, using the thermal cycle method proposed by Goldstein and Reynolds (1994), to measure the hydrocarbon-containing inclusions with stable beating and regular shape and the brine inclusions in the same period the homogeneity temperature (Th) and the freezing point temperature (Tm) of the body were measured.

After being captured, inclusions will be affected by the temperature and pressure of the formation, which will cause deformation and greatly affect the uniform temperature. However, the formation of inclusions is complicated and there are many damage factors. As far as possible, samples consistent with the FIA concept were screened for experimental test analysis to ensure the reliability of experimental data [31,32]. FIA refers to "the most detailed group of related inclusions that can be classified petrographically" or "a group of inclusions that can be distinguished by petrographic methods and represent the most finely divided inclusion capture event" [33,34]. Each FIA is based on a petrographic relationship rather than a similarity in thermometric data representing a temporally finer-grained inclusion storage event [35,36]. The basin simulation software PetroMod was used to draw a burial-thermal history map combined with the uniform temperature of the fluid inclusion body temperature and salt test to determine the time of fluid inclusion capture [37], which provided a basis for correctly dividing the oil and gas migration and charging periods of the Yanchang Formation. This experiment was completed in the National Key Geochemical Laboratory of Yangtze University in Wuhan City, Hubei Province.

## Results and discussion

### Reservoir diagenesis

The diagenesis of the reservoir affects the physical properties of the reservoir, which is the basis for the division of inclusions [38]. The products and traces of diagenesis represent the activity information of deep formation fluids, which are of great significance to oil and gas migration and accumulation [39]. The tight sandstone in the reservoir in the study area is mainly composed of arkose and lithic arkose, with a small amount of medium sandstone and coarse sandstone. Its diagenetic process is complex. Previous studies have shown that the Chang 7 member reservoir in the study area mainly experienced diagenesis including

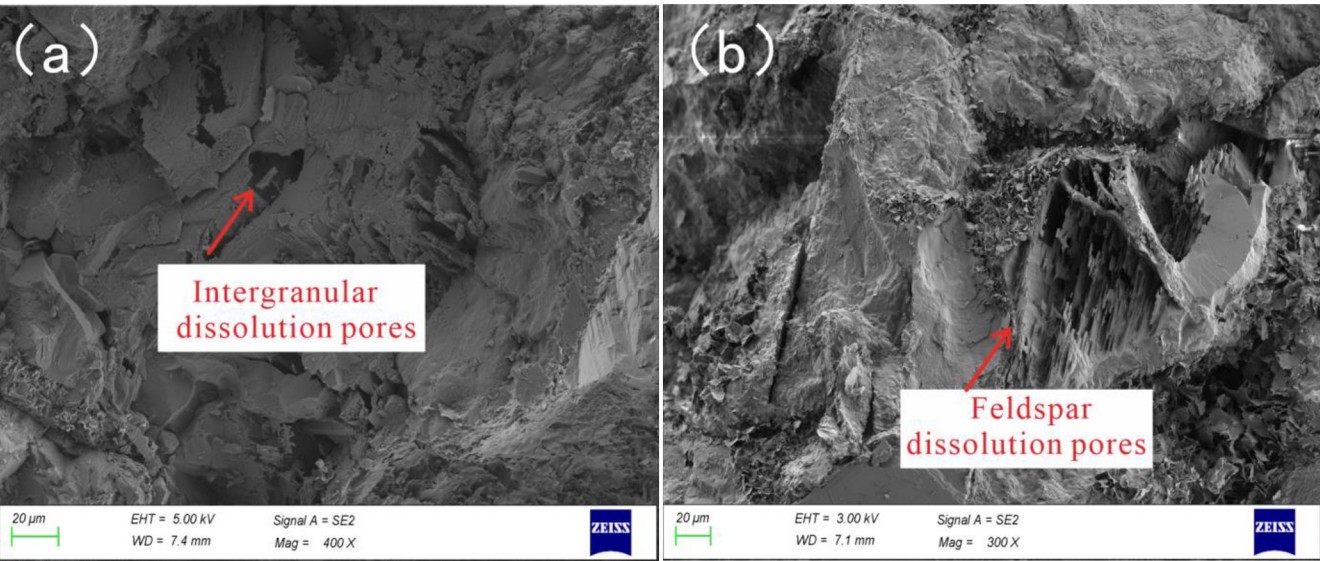

**Fig 2. Diagenetic evolution sequence of Chang 7 member in Zhijing-ansai area.**

compaction, cementation, micro-fractures, and dissolution [40]. The sandstone reservoirs of the Chang 7 member are mainly in the middle diagenetic stage A (Fig 2). According to the study of the burial history, the sandstone of the Chang 7 member has experienced a burial depth of 3000 m, and the compaction is relatively developed. During the early diagenesis stage A, the paleogeothermal temperature was low, the organic matter was immature, and mechanical compaction was the main action, accompanied by early chlorite appearing in the form of film, and a small amount of calcite cement was produced. The authigenic clay minerals in the reservoir are mainly chlorite, illite, kaolinite and illite-smectite mixed layer. In the early diagenetic stage B, the paleogeothermal temperature was 65–85°C, and as the buried depth increased, the compaction gradually increased. At this time, the Ro was 0.35–0.5%, the fluid was weakly acidic, and the feldspar began to dissolve. Its dissolution is mainly the dissolution of clastic particles and the dissolution of interstitials, and the types of pores are residual intergranular pores and secondary dissolved pores. The feldspar dissolution is mainly along the feldspar cleavage seam, forming dissolution pores in the feldspar grains, and the feldspar and cuttings are completely dissolved to form cast film pores (Fig 3). The intergranular dissolution pores and intragranular dissolution pores formed by dissolution make the reservoir pores larger and the physical properties better, which is the most important diagenesis for the development of the Chang 7 member reservoir [41].

## Salt physology observation and composition characteristics of inclusions

The fluid inclusions in the reservoir samples of the Chang 7 member of the Yanchang Formation mainly occur in the quartz grains, mainly in single liquid phase, gas-liquid two-phase, and pure gas phase (Fig 4). Gas-liquid two-phase inclusions include brine inclusions in which bubbles move randomly and gas-rich inclusions (Fig 8).

Through microscope transmitted light and fluorescence observation, blue fluorescent and blue-green fluorescent oils were observed in thin slices of samples collected in Well Xin 283, Well Xin 140, Well Wu 100, Well Shun 111, and Well Shun 37 in the Chang $7_1$ Formation inclusions. It is mainly distributed in quartz particles, the gas-liquid ratio is 3–9%, and the size of a single inclusion is between 5–10 μm. Its shape is mostly irregular round and long strips,

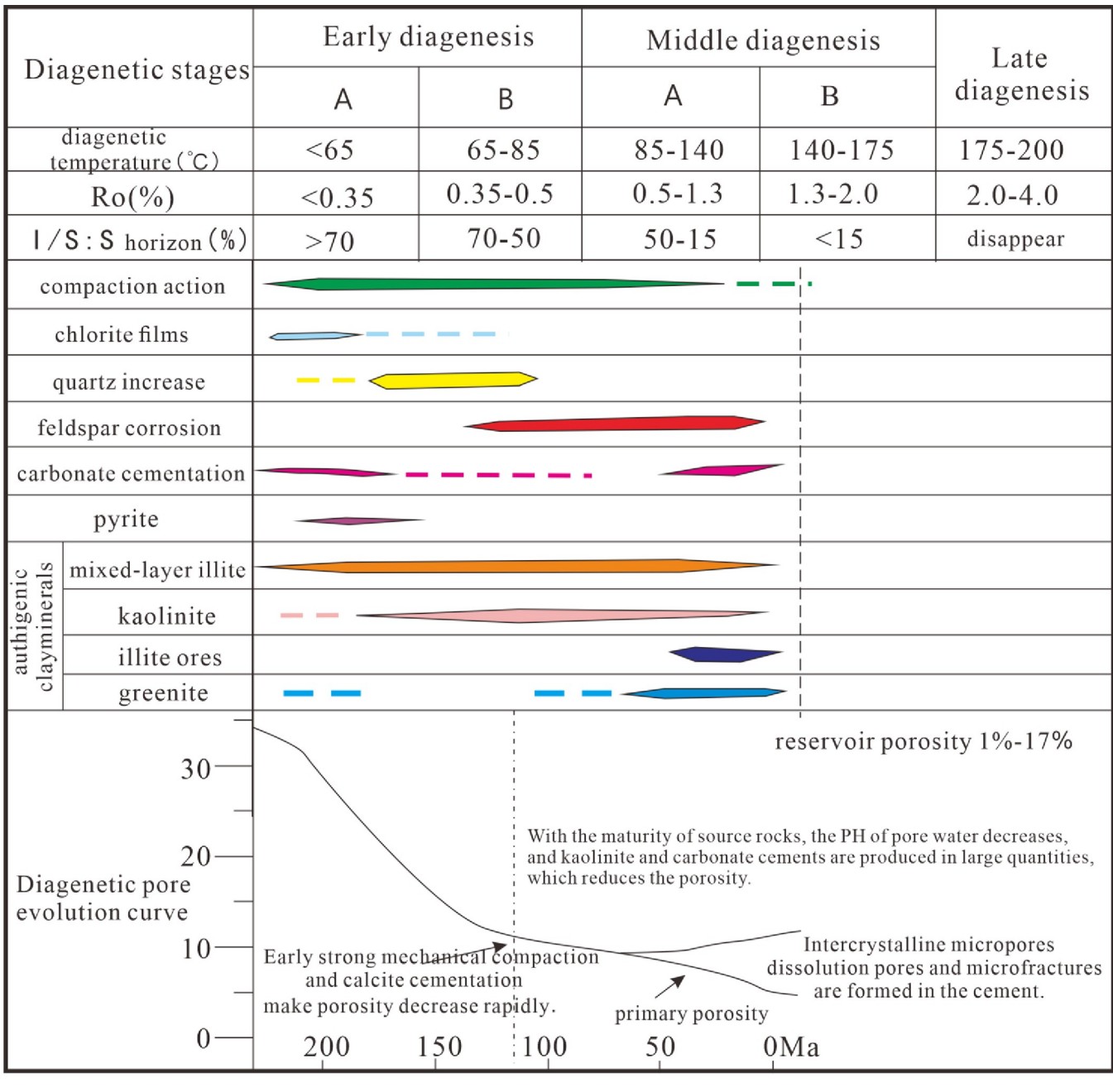

**Fig 3. Characteristics of dissolution pores in Chang 7 member.** (a) Well Qiao136, 1578.25m, intergranular dissolution pores, (b) Well Wu 100, 1937.5m, feldspar dissolution pores.

and most of them are produced in the shape of beads. The reasons for the different fluorescence intensities of inclusions are the differences in maturity and shape of inclusions. The measured oil inclusions in the Chang $7_1$ segment have differences in the peak morphology of the microfluorescence spectrum (Fig 4), and the measured GOI value of the oil inclusions is about 8–15%.

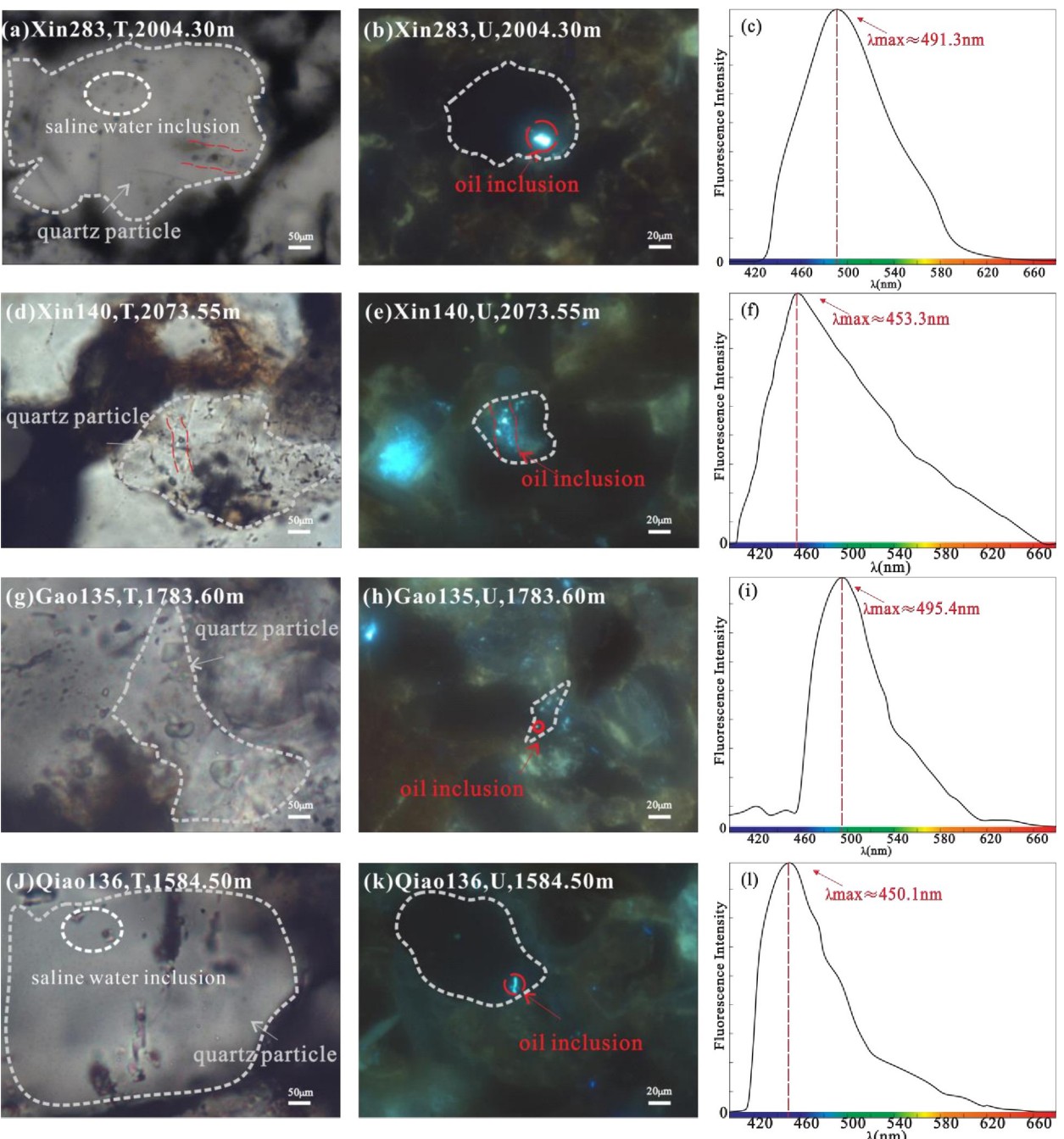

**Fig 4. Fluorescence spectral characteristics of inclusions in Chang 7 member.** (a) Well Xin 283, 2004.30m, liquid hydrocarbon inclusions, 50 times transmitted light. (b) Well Xin 283, 2004.30m, blue-green fluorescence and a in the same field of vision. (c) Well Xin 283, 2004.30m, fluorescence spectra of oil inclusions. (d) Well Xin 140, 2073.55m, Gas-liquid two-phase hydrocarbon inclusions, 50 times transmitted light. (e) Well Xin 140, 2073.55m, blue luminescence. (f) Well Xin 140, 2073.55m, fluorescence spectra of oil inclusions. (g) Well Gao 135, 1783.60m, gas-liquid two-phase hydrocarbon inclusions, 50 times transmitted light. (h) Well Gao 135, 1783.60m, bluish green fluorescence. (i) Well Gao135, 1783.60m, fluorescence spectra of oil inclusions. (j) Well Qiao 136, 1584.50m, liquid hydrocarbon inclusions, 50 times transmitted light. (k) Well Qiao 136, 1584.50m, blue luminescence. (l) Well Qiao136,1584.50m, fluorescence spectra of oil inclusions.

In the thin sections of samples collected from Well Qiao 136 and Well Gao 135 in the Chang $7_2$ member of the Yanchang Formation, it was observed that inclusions are usually developed in mineral fractures and asphaltenes. Most of them are gas-liquid two-phase inclusions, the gas-liquid ratio is 3–9%, and the size of a single inclusion is relatively large, mostly between 7–10 μm (Table 1). The shape is nearly round and rhombus, mostly in single output, and it is light yellow and colorless under single polarized light. Under fluorescence excitation, it shows blue fluorescence with high abundance, and the peak range of the microfluorescence spectrum is 450-491nm.

According to the different phase distribution and composition characteristics of the inclusions, the inclusions measured in the Chang 7 member of the Triassic Yanchang Formation in the study area can be divided into the following four types:

The first type is a single liquid hydrocarbon-containing inclusion, which is mostly taupe under transmitted light, and blue-green or blue under fluorescence. Mainly trapped and fracture-filled in quartz (Fig 4B).

The second type is gaseous hydrocarbon inclusions, which are gray under a single polarizer, and mainly occur in quartz filled with fractures (Fig 8B).

The third type is gas-liquid two-phase salt water inclusions (Fig 4A and 4J), which are transparent under a single polarizer and mostly irregular oval and long strips. It mainly occurs in fracture filling quartz and fracture filling calcite.

The fourth type is bitumen (Fig 8C), dark brown under a single polarizer, no fluorescence display, low abundance, captured in fracture-filling quartz minerals.

According to the different phase distribution and composition characteristics of the inclusions, the inclusions measured in the Chang 7 member of the Triassic Yanchang Formation in the study area can be divided into the following four types:

The first type is a single liquid hydrocarbon-containing inclusion, which is mostly taupe under transmitted light, and blue-green or blue under fluorescence. Mainly trapped and fracture-filled in quartz (Fig 4B).

The second type is gaseous hydrocarbon inclusions, which are gray under a single polarizer, and mainly occur in quartz filled with fractures (Fig 8B).

**Table 1. Statistical table of microscopic temperature measurement of inclusions in Chang 7 member in Zhijing-Ansai area.**

| Well | Layers | Depth /(m) | Number of test inclusions/piece | Size/(μm) | Gas-liquid ratio/% | Temperature/˚C | Freezing point/˚C | GOI/% |
|---|---|---|---|---|---|---|---|---|
| Xin283 | Chang$7_1$ | 2004.3 | 9 | 5–8 | 3–6 | 92.6-105.7 | -3.2–4.1 | 13 |
| Xin140 | Chang$7_1$ | 2073.55 | 8 | 5–7 | 3–7 | 90.1-109.7 | -2.8–3.6 | 8 |
| Qiao136 | Chang$7_2$ | 1584.5 | 11 | 7–9 | 5–8 | 119.3-125.7 | -6.6–2.1 | 11 |
| Gao135 | Chang$7_2$ | 1783.6 | 13 | 7–10 | 6–9 | 121.6-128.9 | -7.7–3.5 | 10 |
| Wu100 | Chang$7_1$ | 1935.2 | 9 | 5–7 | 3–5 | 93.7-106.72 | -6.1–3.6 | 12 |
| Xin283 | Chang$7_1$ | 1995.7 | 8 | 7–9 | 7–8 | 90.8-110.3 | -2.2–3.8 | 9 |
| Shun111 | Chang$7_1$ | 1756.3 | 9 | 5–10 | 6–9 | 91.2-103.1 | -1.9–3.3 | 13 |
| Shun37 | Chang$7_1$ | 1917.25 | 10 | 6–8 | 5–8 | 91.3-107.6 | -3.5–3.8 | 10 |

The third type is gas-liquid two-phase salt water inclusions (Fig 4A and 4J), which are transparent under a single polarizer and mostly irregular oval and long strips. It mainly occurs in fracture filling quartz and fracture filling calcite.

The fourth type is bitumen (Fig 8C), dark brown under a single polarizer, no fluorescence display, low abundance, captured in fracture-filling quartz minerals.

## Thermometric characteristics of inclusions and oil and gas accumulation and charging stages

Inclusion microthermometry experiments were carried out in the Chang 7 member of the Yanchang Formation, and the results showed that: Well Xin 283, Xin 140, Shun 111, Wu 100 and Gao 135, The inclusions developed in Well Qiao 136 have similar physical properties. The FIA representing the two-stage oil and gas charging process was detected, and the detection results proved that there were two stages of oil and gas charging in the Chang 7 member in the Zhijing-Ansai area. The detected oil inclusions with two fluorescent characteristics are generally distributed in diagenetic fractures within quartz grains, which corresponds to early capture events. Due to the different structural positions, there are certain differences in the burial history-thermal history evolution of single wells, resulting in a certain degree of difference in the homogeneity temperature of the associated brine inclusions of oil inclusions. The homogeneous temperature of brine inclusions in the first stage is distributed between 90–110˚C, with an average of 106.8˚C, the homogeneous temperature of brine inclusions in the second stage is mainly distributed between 120–130˚C, with an average of 126.7˚C (Fig 5).

The salinity value of the fluid inclusions is indirectly obtained from the measurement of the freezing point temperature of the inclusions, and the salinity value of the fluid inclusions is between 5.35–13.06%. The intersection graph of the homogeneous temperature and salinity of the inclusions shows that there is no obvious correlation between the homogeneous temperature of the inclusions in the two stages and the salinity (Fig 6).

According to the burial thermal evolution history of the Chang 7 member and the measured homogeneous temperature of the associated brine inclusions in the same period, the oil and gas charging time was estimated. The results show that the Chang 7 member is mainly formed in two phases. Phase I oil and gas inclusions are the period of large-scale oil and gas

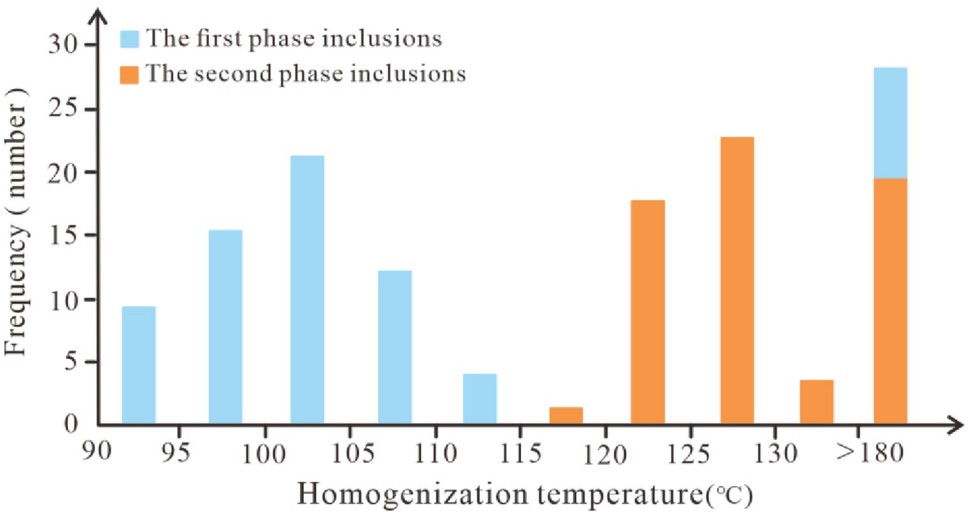

**Fig 5. Histogram of homogeneous temperature distribution of inclusions in Chang 7 member.**

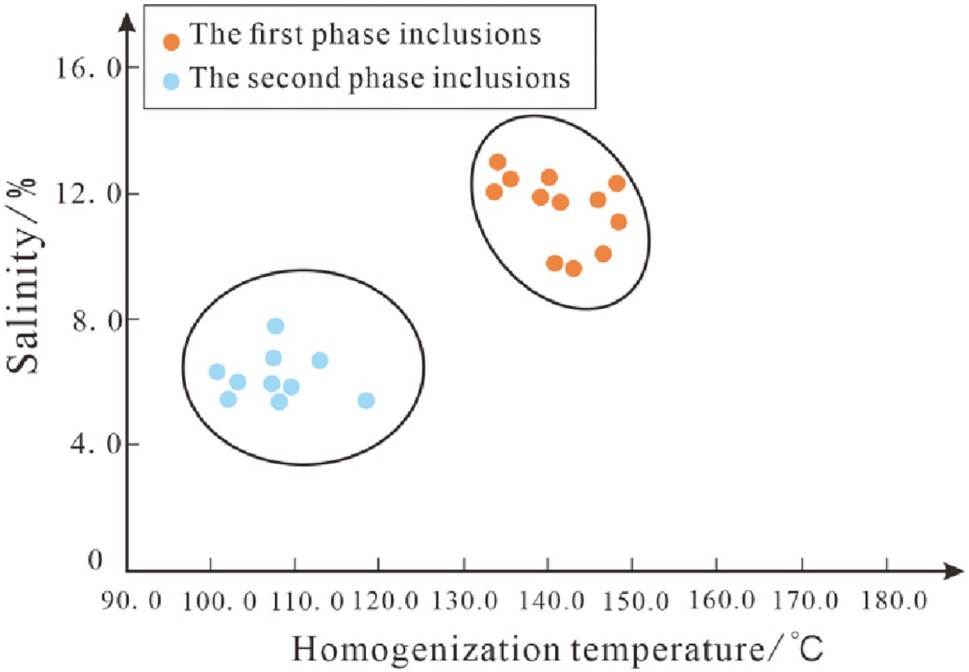

**Fig 6. Scatter diagram of homogeneous temperature-salinity of inclusions in Chang 7 member.**

accumulation. Combined with the homogeneous temperature of brine inclusions in this period of oil inclusions in the same period and the simulation results of thermal history restoration, it is judged that the oil and gas charging period corresponds to about 120–100 Ma ago, which is the Early Cretaceous late stage (Fig 7). In the second stage of accumulation, the abundance of oil and gas inclusions in the stage is relatively high. Combined with the homogeneous temperature of brine inclusions associated with oil and gas inclusions in this period and the

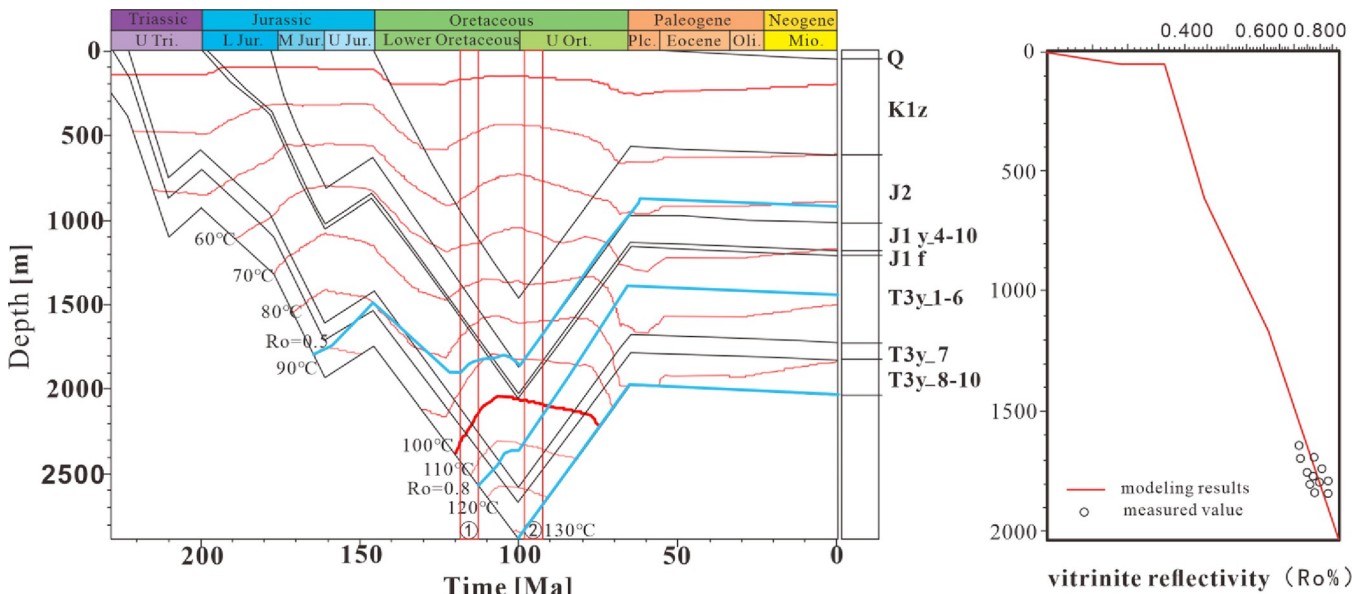

**Fig 7. Burial history-thermal restoration map of Mesozoic Triassic Yanchang Formation in Zhijing-Ansai area, Ordos Basin.**

paleogeothermal history of the reservoir, the corresponding time is about 100–90 Ma, which means that the second oil and gas charge was in the early Late Cretaceous (Fig 7). Since the Late Triassic-Early Cretaceous was the formation stage of the Ordos Inland Basin, the platform subsidence amplitude was roughly the same, and the measured oil and gas charging times of the two periods were close.

## Discussion on the source of $CO_2$ inclusions and high temperature inclusions

While hydrocarbon-related inclusions were observed in the Chang 7 member reservoir, gas-rich phase inclusions, $CO_2$ inclusions, and high-temperature inclusions were also observed. It is mostly trapped in the quartz minerals with developed fractures.The samples of $CO_2$ inclusions and high-temperature inclusions in brine in the same period were tested, and it was found that the freezing point ranged from -5.2 to 6˚C, and the uniform temperature was greater than 180˚C (Fig 8).

The sources of $CO_2$ in crustal-surface sedimentary fluids and natural gas are mainly divided into organic sources and inorganic sources [42–44]. The inorganic origin can be divided into mantle origin and petrochemical origin. One is that volcanic activity is accompanied by the eruption of a large amount of high-temperature gas [45]. During the up welling of magma, the temperature and pressure drop, and the carbon dioxide it carries is released. The second is formed by the decomposition of calcium carbonate in the formation under the high temperature action of magma or thermal fluid [46].

The main source of carbon dioxide formation in the Ordos Basin is related to the late Paleozoic volcanic activities [47]. Extensive volcanic activity occurred during this period, and magmatic fluids associated with volcanic activity may contain dissolved carbon dioxide. This carbon dioxide can be released during volcanic eruptions, or it can be transferred to the sedimentary rocks around the basin through fractures, pores, caves, unconformities and faults [48].

As far as the Yanchang Formation is concerned, $CO_2$ inclusions can come from $CO_2$-rich fluids or deep formations. $CO_2$ in the strata can migrate vertically through different geological structures [49]. Thermal fluids can transport carbon dioxide from deeper formations to shallower formations through fractures and faults as a medium. As the hot fluid continues to rise and the pressure gradually decreases, $CO_2$ dissolves from the magma, forming gas bubbles, some of which can be trapped in the volcanic rock or released into the surrounding environment. The burial and heating of these volcanic rocks resulted in the release of $CO_2$ as a by-product, leading to the formation of $CO_2$ inclusions [50].

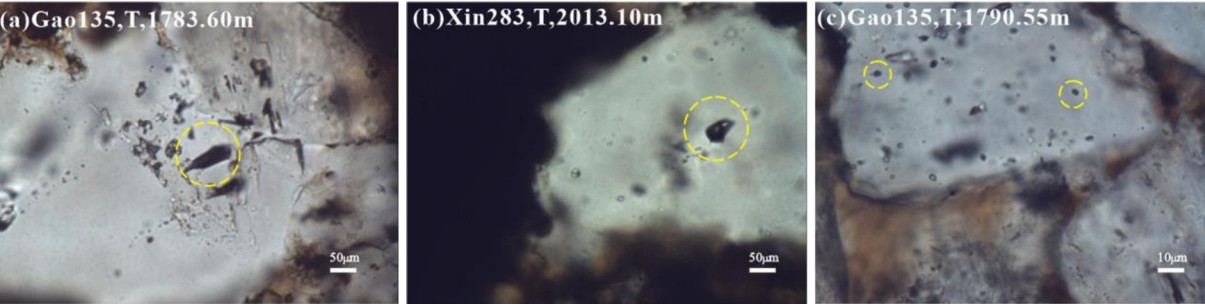

**Fig 8. Numerous gas, $CO_2$ and asphalt inclusions in Chang 7 member.** (a) Well Gao 135, 1783.60m, 50 times transmitted light, rich gas phase inclusions. (b) Well Xin 283, 2013.10m, 50 times transmitted light, $CO_2$ inclusions. (c) Well Gao135, 1790.55m high, 10 times transmitted light, asphalt.

According to Ren et al. (2020), a tectonic thermal event occurred during the Early Cretaceous. The large-scale generation and accumulation period of oil and gas accumulation in the Ordos Basin was in the Early Cretaceous [51]. The tectonic thermal events in the Early Cretaceous controlled the oil and gas generation and accumulation periods of the main source rocks in the Mesozoic, which corresponds to the accumulation time of the inclusion test analysis. It is speculated that the source of $CO_2$ inclusions and high temperature inclusions in the Chang 7 member of the Mesozoic Yanchang Formation in the Zhijing-Ansai area originated from the up welling of as the no spheric materials in the deep Ordos Basin in the Early Cretaceous. It is speculated that the $CO_2$ inclusions and high-temperature inclusions in the Chang 7 member in the Zhijing-Ansai area originated from the up welling of deep as the no spheric materials in the Ordos Basin during the Early Cretaceous. The specific formation reasons and the accumulation time of $CO_2$ inclusions need further experimental research.

## Conclusions

The fluid inclusions in the Chang 7 member reservoir of the Mesozoic Triassic Yanchang Formation in the Zhijing-Ansai area of the Ordos Basin are mainly gas-liquid two-phase brine inclusions and liquid hydrocarbon inclusions. It also includes a small amount of gaseous hydrocarbon inclusions and asphalt, and the main component of gaseous hydrocarbon inclusions is $CO_2$.

The fluid inclusions developed in the reservoirs of the Mesozoic Triassic Yanchang Formation in the Zhijing-Ansai area can be divided into two stages. Among them, the phase I inclusions are mainly liquid hydrocarbon inclusions and a large number of associated brine inclusions, mainly showing blue and blue-green fluorescence. Predominantly beaded distribution of early fracture-filled quartz and fracture-filled calcite. The peak homogeneity temperature of its associated brine inclusions is between 90–110˚C. Phase II inclusions mainly develop gas-liquid two-phase hydrocarbon inclusions, associated brine inclusions and a small amount of $CO_2$ inclusions and bitumen. It mostly occurs in the form of bands in fracture-filled quartz, and the peak homogeneity temperature of its associated brine inclusions is between 120–130˚C.

The Chang 7 reservoir of the Yanchang Formation in the Ordos Basin mainly experienced two stages of oil and gas charging. The first period occurred in the late Early Cretaceous. It is a period of massive oil and gas charging, and it is the period of oil and gas generation and accumulation of main source rocks. The second charge occurred in the early Late Cretaceous.

The formation of $CO_2$ inclusions and high temperature inclusions in Chang 7 member of Zhijing-Ansai area is related to the late Paleozoic volcanic activity, which is mainly derived from the up welling of deep as the no sphere in the Ordos Basin in the early Cretaceous.

## Author Contributions

**Conceptualization:** Wenjie Wu.

**Data curation:** Yong Feng, Yulin Chen.

**Formal analysis:** Wenjie Wu, Nan Wu, Yilin Liang.

**Funding acquisition:** Jian Wang, Nan Wu, Yilin Liang.

**Investigation:** Wenjie Wu, Jian Wang, Nan Wu.

**Methodology:** Wenjie Wu, Jian Wang, Nan Wu.

**Project administration:** Jian Wang, Nan Wu.

**Resources:** Wenjie Wu, Jian Wang.

**Visualization:** Yong Feng.

**Writing – original draft:** Wenjie Wu.

**Writing – review & editing:** Wenjie Wu.

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
