## [Decision Letter · Decision Letter 0]

10 Dec 2023

PONE-D-23-39049Characteristics of Inclusions in Chang 7 Member and Shale Oil Accumulation Stages in Zhijing-Ansai Area, Ordos BasinPLOS ONE

Dear Dr. Wang,

Thank you for submitting your manuscript to PLOS ONE. After careful consideration, we feel that it has merit but does not fully meet PLOS ONE’s publication criteria as it currently stands. Therefore, we invite you to submit a revised version of the manuscript that addresses the points raised during the review process.

I strongly ask the authors to fully follow the reviewer suggestions, including the suggested improvement of the cited references that are, in the present form, too much local. Please consider current worldwide literature. 

We look forward to receiving your revised manuscript.

Kind regards,

Fabio Trippetta, Ph.D.

Academic Editor

PLOS ONE

Journal Requirements:

4. Please note that funding information should not appear in any section or other areas of your manuscript. We will only publish funding information present in the Funding Statement section of the online submission form. Please remove any funding-related text from the manuscript.

6. Thank you for stating the following financial disclosure: 

   " the National Nature Science Foundation of China"

7. Thank you for stating the following in your Competing Interests section:  

   "NO"

8. In your Data Availability statement, you have not specified where the minimal data set underlying the results described in your manuscript can be found. PLOS defines a study's minimal data set as the underlying data used to reach the conclusions drawn in the manuscript and any additional data required to replicate the reported study findings in their entirety. All PLOS journals require that the minimal data set be made fully available. For more information about our data policy, please see http://journals.plos.org/plosone/s/data-availability.

9. We note that you have stated that you will provide repository information for your data at acceptance. Should your manuscript be accepted for publication, we will hold it until you provide the relevant accession numbers or DOIs necessary to access your data. If you wish to make changes to your Data Availability statement, please describe these changes in your cover letter and we will update your Data Availability statement to reflect the information you provide.

10. PLOS requires an ORCID iD for the corresponding author in Editorial Manager on papers submitted after December 6th, 2016. Please ensure that you have an ORCID iD and that it is validated in Editorial Manager. To do this, go to ‘Update my Information’ (in the upper left-hand corner of the main menu), and click on the Fetch/Validate link next to the ORCID field. This will take you to the ORCID site and allow you to create a new iD or authenticate a pre-existing iD in Editorial Manager. Please see the following video for instructions on linking an ORCID iD to your Editorial Manager account: https://www.youtube.com/watch?v=_xcclfuvtxQ

11. We note that Figure 1 in your submission contain map/satellite images which may be copyrighted. All PLOS content is published under the Creative Commons Attribution License (CC BY 4.0), which means that the manuscript, images, and Supporting Information files will be freely available online, and any third party is permitted to access, download, copy, distribute, and use these materials in any way, even commercially, with proper attribution. For these reasons, we cannot publish previously copyrighted maps or satellite images created using proprietary data, such as Google software (Google Maps, Street View, and Earth). For more information, see our copyright guidelines: http://journals.plos.org/plosone/s/licenses-and-copyright.

Reviewers' comments:

Reviewer's Responses to Questions

**Comments to the Author**

1. Is the manuscript technically sound, and do the data support the conclusions?

Reviewer #1: Yes

Reviewer #2: Yes

2. Has the statistical analysis been performed appropriately and rigorously? 

Reviewer #1: Yes

Reviewer #2: Yes

3. Have the authors made all data underlying the findings in their manuscript fully available?

Reviewer #1: Yes

Reviewer #2: Yes

4. Is the manuscript presented in an intelligible fashion and written in standard English?

Reviewer #1: Yes

Reviewer #2: Yes

5. Review Comments to the Author

Reviewer #1: The paper analyzes fluid inclusions in reservoir samples from the Chang 7 member of the Yanchang Formation in the Ordos Basin using microscopic observation, temperature measurement, and fluorescence spectroscopy.

It identifies two stages of oil and gas accumulation based on the characteristics of fluid inclusions, including inclusion type, temperature, and fluorescence. The first stage occurred in the late Early Cretaceous and the second in the early Late Cretaceous.

It discusses the sources of CO2 inclusions and high-temperature inclusions, relating them to volcanic activity in the Early Cretaceous.

The study helps understand the oil and gas accumulation mechanisms and periods in the Yanchang Formation to guide exploration in the area.

The study uses multiple well-established petrographic techniques to analyze fluid inclusions and comprehensively interpret the oil and gas accumulation history.

It relates the inclusion data to burial history modeling and tectonic events to constrain accumulation timings.

The findings provide insights on the key accumulation periods to inform exploration in the region.

Questions for Authors:

What criteria were used to select inclusions for analysis (e.g. inclusion size, shape)?

How was the microscope stage calibrated and what was the estimated temperature uncertainty?

Have other inclusion studies in the area reported consistent or differing accumulation timings? If differing, how do the authors reconcile the interpretations?

The study presents a comprehensive inclusion analysis and ties the findings to the geological evolution of the area. While some methodological details could be improved, it provides useful insights for the basin. The questions aim to strengthen the reliability and contextualize the conclusions.

Reviewer #2: The topic of this article is relatively new, and the views are clear and correct. The article conforms to the purpose of this journal, and also reflects strong innovation and practical application. The structure of the full text is basically reasonable, the ideas are clear, and the levels are clear. The expression of opinions is basically accurate, and the arguments and arguments are basically consistent.

This article requires the following changes from the author:

1. The referenced literature needs to be closely integrated with the topic and paper materials, and it is recommended to use newer references.

2. It is recommended to supplement the experimental data in Table 1.

3. It is recommended that a scale bar be added to Figure 8.

4. The title "chang 7" in Figure 6 is not standard and should be changed to "chang 7 member".

5. The marked range in Figure 4 is not clear, replace it with an obvious outline.

6. Finally, pay attention to the writing format of experimental data in the article.

6. PLOS authors have the option to publish the peer review history of their article (what does this mean?). If published, this will include your full peer review and any attached files.

Reviewer #1: **Yes: **Xu-Guang Guo

Reviewer #2: **Yes: **Wei Lin

---

## [Author Response · Author response to Decision Letter 0]

18 Mar 2024

Responses to Editor

I have received your email, for your proposed amendments, I am deeply grateful. I have modified according to your opinion, and submitted the modified paper in the journal paper system.

If you have any questions, please feel free to contact me. I will address them promptly and make the necessary corrections.

Responses to Reviewer 1

Thank you for your comments, we have studied the valuable comments from you and 

tried our best to revise the manuscript. All the modified parts are marked up using 

“track changes” fuction of Word. The details are as follows:

1.Comment:What criteria were used to select inclusions for analysis (e.g. inclusion size, shape)?

Response:Thank you for your question. According to the People 's Republic of China oil and gas industry standard SY /T 6010-1994 sedimentary rock inclusion homogenization temperature and salinity determination method, the inclusion test was carried out. In theory, any mineral crystallized from the fluid will contain inclusions. The main mineral ( quartz ) grains with good transparency and good crystallinity were selected for microscopic observation from low to high magnification.

2.Comment:How was the microscope stage calibrated and what was the estimated temperature uncertainty?

Response:Thank you for your question. Firstly, the microscope is placed under the light source, and the brightness and uniformity of the light source are adjusted to make the observation area have enough light and appropriate contrast. The second step is to adjust the eyepiece. Next, adjust the focal length of the eyepiece, so that the field of view of the eyepiece is clear and without distortion. It can be achieved by rotating the focusing wheel of the eyepiece. The third step is to adjust the objective lens : select the appropriate objective lens, adjust the focal length of the objective lens, so that the image of the observed sample is clearly visible. Similarly, it can be achieved by rotating the focusing wheel of the objective lens. Step 4 Calibration scale : In the field of view of the microscope, there is usually a scale to measure the length or diameter of the sample. First, a standard sample with a known length is measured under a microscope, and then the scale of the microscope is adjusted to match the standard length. The fifth step of calibrating the eyepiece micrometer is to use a standard sample with a known length to measure under the microscope, and then adjust the scale of the eyepiece micrometer to make it consistent with the standard length. Finally, check the microscope 's focusing system to ensure that clear images can be obtained under different observation conditions. The measurement error of the micro cold and hot stage instrument is ± 0.1 °C.

3.Comment:Have other inclusion studies in the area reported consistent or differing accumulation timings? If differing, how do the authors reconcile the interpretations?

Response:Thank you for your question. Other scholars in the Yanchang Formation of the Ordos Basin have studied the accumulation period of inclusions. The reservoir accumulation period of Mesozoic Yanchang Formation in southeastern Ordos Basin is mainly Early Cretaceous ( 93-120Ma ). This paper is aimed at the accumulation period of the Yanchang Formation in Zhijing Ansai area. Although it is affected by the region, the recovery of the accumulation period echoes with previous scholars.

Responses to Reviewer 2

Thank you for your comments, we have studied the valuable comments from you and 

tried our best to revise the manuscript. All the modified parts are marked up using 

“track changes” fuction of Word. The details are as follows:

1.Comment:The referenced literature needs to be closely integrated with the topic and paper materials, and it is recommended to use newer references.

Response:Thank you for your suggestions. I have made modifications to the questions raised and highlighted them.

2.Comment:It is recommended to supplement the experimental data in Table 1.

Response:Thanks for your valuable advice. All relevant data are within the paper.

3.Comment:It is recommended that a scale bar be added to Figure 8.

Response:Thank you for your question. I will make modifications to Figure 8 and highlight it in the manuscript.

4.Comment:The title "chang 7" in Figure 6 is not standard and should be changed to "chang 7 member".

Response:Thanks for pointing out the error.I have made corrections to this issue.

5.Comment:The marked range in Figure 4 is not clear, replace it with an obvious outline.

Response:Thanks for your valuable advice.I have made corrections to the images, please refer to the manuscript.

6.Comment:Finally, pay attention to the writing format of experimental data in the article.

Response:Thank you for pointing out the error. I have made modifications to the format.

---

## [Decision Letter · Decision Letter 1]

4 Apr 2024

Characteristics of Inclusions in Chang 7 Member and Shale Oil Accumulation Stages in Zhijing-Ansai Area, Ordos Basin

PONE-D-23-39049R1

Dear Dr. Wang,

We’re pleased to inform you that your manuscript has been judged scientifically suitable for publication and will be formally accepted for publication once it meets all outstanding technical requirements.

Kind regards,

Fabio Trippetta, Ph.D.

Academic Editor

PLOS ONE

Reviewers' comments:

Reviewer's Responses to Questions

**Comments to the Author**

1. If the authors have adequately addressed your comments raised in a previous round of review and you feel that this manuscript is now acceptable for publication, you may indicate that here to bypass the “Comments to the Author” section, enter your conflict of interest statement in the “Confidential to Editor” section, and submit your "Accept" recommendation.

Reviewer #2: All comments have been addressed

2. Is the manuscript technically sound, and do the data support the conclusions?

Reviewer #2: Yes

3. Has the statistical analysis been performed appropriately and rigorously? 

Reviewer #2: Yes

4. Have the authors made all data underlying the findings in their manuscript fully available?

Reviewer #2: Yes

5. Is the manuscript presented in an intelligible fashion and written in standard English?

Reviewer #2: Yes

6. Review Comments to the Author

Reviewer #2: The revised manuscript is generally satisfactory, and most of my concerns have been well addressed, my recommendation is for acceptance.

7. PLOS authors have the option to publish the peer review history of their article (what does this mean?). If published, this will include your full peer review and any attached files.

Reviewer #2: No
